# Precise Regret Bounds for Log-loss via a Truncated Bayesian Algorithm

**Changlong Wu**[1]    **Mohsen Heidari**[1,2]    **Ananth Grama**[1]    **Wojciech Szpankowski**[1]
[1]CSoI, Purdue University    [2] Indiana University
wuchangl@hawaii.edu, {mheidari,ayg,szpan}@purdue.edu

## Abstract

We study sequential general online regression, known also as sequential probability assignments, under logarithmic loss when compared against a broad class of experts. We obtain tight, often matching, lower and upper bounds for sequential minimax regret, which is defined as the excess loss incurred by the predictor over the best expert in the class. After proving a general upper bound we consider some specific classes of experts from Lipschitz class to bounded Hessian class and derive matching lower and upper bounds with provably optimal constants. Our bounds work for a wide range of values of the data dimension and the number of rounds. To derive lower bounds, we use tools from information theory (e.g., Shtarkov sum), and for upper bounds we resort to new "smooth truncated covering" of the class of experts. This allows us to find constructive proofs by applying a simple and novel truncated Bayesian algorithm. Our proofs are substantially simpler than the existing ones and yet provide tighter (and often optimal) bounds.

## 1 Introduction

In online learning and sequential probability assignments arising in information theory, portfolio optimization, and machine learning, the training algorithm consumes $d$ dimensional data in rounds $t \in \{1, 2, \ldots, T\}$ and predicts the label $\hat{y}_t$ based on data received and labels observed so far. After prediction, the true label $y_t$ is revealed and the loss $\ell(y_t, \hat{y}_t)$ is incurred. The (pointwise) *regret* is defined as the (excess) loss incurred by the algorithm over a class of experts, also called the hypothesis class.

More precisely, in each round $t \geq 1$ the learner obtains a $d$ dimensional input/ feature vector $\mathbf{x}_t \in \mathbb{R}^d$. In addition to $\mathbf{x}_t$, the learner may use the past observations $(\mathbf{x}_r, y_r)$, $r < t$ to make a prediction $\hat{y}_t$ of true label. Therefore, the prediction can be written as $\hat{y}_t = \phi_t(y^{t-1}, \mathbf{x}^t)$, where $y^{t-1}$ represents the labels in the past $t - 1$ rounds, $\mathbf{x}^t$ represents the input vectors in $t$ rounds, and $\phi_t$ represents the strategy of the learner to obtain its prediction based on the past and current observations. Once a prediction is made, nature reveals the true label $y_t$ and the learner incurs loss $\ell : \hat{\mathcal{Y}} \times \mathcal{Y} \to \mathbb{R}$, where $\hat{\mathcal{Y}}$ and $\mathcal{Y}$ are the prediction and true label domains respectively. Hereafter, we assume throughout $\hat{\mathcal{Y}} = [0, 1]$, $\mathcal{Y} = \{0, 1\}$ with logarithmic loss

$$\ell(\hat{y}_t, y_t) = -y_t \log(\hat{y}_t) - (1 - y_t) \log(1 - \hat{y}_t). \tag{1}$$

In regret analysis, we are interested in comparing the accumulated loss of the learner with that of the best strategy within a predefined class of predictors (experts) denoted by $\mathcal{H}$. More precisely, $\mathcal{H}$ is a collection of predicting functions $h : \mathbb{R}^d \mapsto \hat{\mathcal{Y}}$ with input being $\mathbf{x}_t$ at time $t$. Therefore, given a learner $\phi_t, t > 0$ and $(y_t, \mathbf{x}_t)_{t=1}^T$ after $T$ rounds the *pointwise regret* is defined as

$$R(\phi^T, y^T, \mathcal{H} | \mathbf{x}^T) = \sum_{t=1}^T \ell(\hat{y}_t, y_t) - \inf_{h \in \mathcal{H}} \sum_{t=1}^T \ell(h(\mathbf{x}_t), y_t),$$

36th Conference on Neural Information Processing Systems (NeurIPS 2022).

where $\hat{y}_t = \phi_t(y^{t-1}, \mathbf{x}^t)$. Observe that the first term above represents the accumulated loss incurred by the learning algorithm, while the second summation deals with the best prediction within $\mathcal{H}$. We highlight two useful perspectives on analyzing the regret next.

**Fixed Design:** This point of view studies the minimal regret for the worst realization of the label with the feature vector $\mathbf{x}^T$ known in advance. Suppose that the learner has a fixed strategy $\phi_t, t > 0$. Then, the *fixed design minimax regret* for a given $\mathbf{x}^T$ is defined as

$$r_T^*(\mathcal{H}|\mathbf{x}^T) = \inf_{\phi^T} \sup_{y^T} R(\phi^T, y^T, \mathcal{H}|\mathbf{x}^T). \tag{2}$$

Further, the fixed design *maximal* minimax regret is given by:

$$r_T^*(\mathcal{H}) = \sup_{\mathbf{x}^T} \inf_{\phi^T} \sup_{y^T} R(\phi^T, y^T, \mathcal{H}|\mathbf{x}^T). \tag{3}$$

**Sequential Design:** In this paper we mostly focus on the *sequential* or *agnostic* regret in which the optimization on regret is performed at each time $t$ without knowing in advance $\mathbf{x}^T$ or $y^T$. Then the *sequential (maximal) minimax regret* is given by [24]:

$$r_T^a(\mathcal{H}) = \sup_{\mathbf{x}_1} \inf_{\hat{y}_1} \sup_{y_1} \cdots \sup_{\mathbf{x}_T} \inf_{\hat{y}_T} \sup_{y_T} R(\hat{y}^T, y^T, \mathcal{H}|\mathbf{x}^T). \tag{4}$$

In [37] it is shown that $r_T^a(\mathcal{H}) \geq r_T^*(\mathcal{H})$ for all $\mathcal{H}$. We will use $r_T^*(\mathcal{H})$ as our tool to derive lower bounds for $r_T^a(\mathcal{H})$.

Our main goal is to gain insights into the growth of sequential regret $r_T^a(\mathcal{H})$ for various classes $\mathcal{H}$ and to show how the structure of $\mathcal{H}$, as well as the relationship between $d$ and $T$ impact the precise growth of the regret. To see this more clearly, we briefly review regret in universal source coding.

**Regrets in Information Theory.** In universal compression, the dependence between regret and the reference class was intensively studied [10, 21, 27, 28, 32, 38, 39]. Here, there is no feature vector $\mathbf{x}^t$, and the dimension $d = 1$. A sequence $y^T$ is generated by a source $P$ that belongs to a class of sources $\mathcal{S}$, which can be viewed as the reference class $\mathcal{H}$ in online learning. The minimax regret for the logarithmic loss is given by [9, 31, 10]:

$$r_T^*(\mathcal{S}) = \min_Q \max_{y^T}[-\log Q(y^T) + \log \sup_{P \in \mathcal{S}} P(y^T)],$$

where $Q$ is the universal probability assignment approximating the unknown $P$. The main question is how the structure of $\mathcal{S}$ impacts the growth of the minimax regret. Let $m$ denote the alphabet size (in online learning, we only consider $m = 2$). It is known [10, 21, 27, 28, 32, 38, 39] that for Markov sources of order $r$, regret grows as $m^r(m-1)/2 \log T$ for fixed $m$ [27, 21, 28, 33], while in [33] minimax regret was analyzed for all ranges of $m$ and $T$. For non-Markovian sources, the growth is super logarithmic. For example, for renewal sources of order $r$ the regret is $\Theta(T^{r/(r+1)})$ [7] and the precise constant in front of the leading term is know for $r = 1$ [11]. However, it should be pointed out that [6, 1] studied the general classes of densities smoothly parameterized by a $d$-dimensional data to obtain general results for minimax regret that can be phrased as an online regret.

**Main Contributions.** Our main results are summarized in Table 1. One of the main contributions of this paper is the concept of a global sequential covering used to prove *constructively* general upper bounds on regret (Theorem 1). We establish Theorem 1 via a novel smooth truncation approach enabling us to find tight upper bounds that subsume the state-of-the-art results (e.g., [23, 3]) obtained non-algorithmically. In fact, Algorithm 2 developed in this paper achieves these bounds. Moreover, Theorem 1 provides optimal constants that are crucial to derive the best bounds in special cases discussed next. For general Lipschitz parametric class $\mathcal{H}$, in Theorem 2, we derive the upper bound $d \log(T/d) + O(d)$ for $T > d$. In Theorem 3, we show that the leading constant 1 (in front of $d \log(T/d)$) is optimal for $T \gg d \log(T)$. Furthermore, we obtain the best constant for the leading term $\frac{d}{2} \log(T/d)$ when the Hessian of $\log f$ is bounded for any function $f \in \mathcal{H}$ (see Theorem 4). Then, we show in Theorem 5 that the constant $\frac{1}{2}$ in our bound is optimal for functions of the form $f(\langle \mathbf{w}, \mathbf{x} \rangle)$, where $\mathbf{w} \in \mathbb{R}^d$ is the parameter of the function, $\langle \mathbf{w}, \mathbf{x} \rangle$ is the inner product in $\mathbb{R}^d$, $\mathbf{w}$ and $\mathbf{x}$ are in a general $\ell_s$-norm unite ball, and $T \gg d^{(s+2)/s}$. This result recovers all the lower bounds in [29] obtained for logistic regression (however, the technique of [29] works for other functions with

bounded second derivatives, like the probit function). Lastly, when $d \geq T$, for a linear function of the form $|\langle \mathbf{w}, \mathbf{x} \rangle|$ we show that the growth is at least $\Omega(T^{s/(s+1)})$ under $\ell_s$ ball and at most $\tilde{O}(T^{2/3})$ for $\ell_2$ ball (see Theorem 6 and Example 2).

The main technique used in our paper (smooth truncation) is novel with other potential applications (e.g., average minimax regret). Instead of the conventional approach for truncating only the values close to $\{0, 1\}$, we truncate all values in $[0, 1]$ in a smooth way (see Lemma 4). This allows us to obtain an upper bound via a simple truncated Bayesian algorithm. Our proofs are substantially simpler (cf. [3]) yet provide tighter and often optimal bounds.

In summary, our main contributions are: (i) constructive proofs through a new smooth truncated Bayesian algorithm; (ii) the novel application of global sequential covering in the context of logarithmic loss; (iii) lower and upper bounds with optimal leading constants; and (iv) novel information-theoretic techniques for the lower bounds.

Table 1: Summary of results

| **Constrains** | $d$ v.s $T$ | **Bounds** | **Comment** |
|---|---|---|---|
| General $\alpha$ cover $\mathcal{G}_\alpha$ | N.A. | $r_T^a(\mathcal{H}) \leq \inf\limits_{0 < \alpha < 1} \{2\alpha T + \log |\mathcal{G}_\alpha|\}$ | Theorem 1 |
| General Lipschitz $f$ | Any | $r_T^a(\mathcal{H}_f) \leq d \log\left(\frac{T}{d} + 1\right) + O(d)$ | Theorem 2 |
| under $\ell_s$ ball | $T \gg d \log T$ | $r_T^a(\mathcal{H}_f) \geq d \log\left(\frac{T}{d}\right) - O(d \log \log T)$ | Theorem 3 |
| Bounded Hessian of $\log f$ under $\ell_2$ ball | Any | $r_T^a(\mathcal{H}_f) \leq \dfrac{d}{2} \log\left(\dfrac{T}{d} + 1\right) + O(d)$ | Theorem 4 |
| $f(\langle \mathbf{w}, \mathbf{x} \rangle)$ with $f'(0) \neq 0$ under $\ell_s$ ball | $T \gg d^{(s+2)/s}$ | $r_T^a(\mathcal{H}_f) \geq \frac{d}{2} \log\left(\frac{T}{d^{(s+2)/s}}\right) - O(d)$ | Theorem 5 |
| $|\langle \mathbf{w}, \mathbf{x} \rangle|$ under $\ell_s$ ball | $d \geq T$ | $r_T^a(\mathcal{H}_f) \geq \frac{s+1}{s \cdot e} T^{s/(s+1)}$ | Theorem 6 |
| $|\langle \mathbf{w}, \mathbf{x} \rangle|$ under $\ell_2$ ball | $d \geq T$ | $\Omega(T^{2/3}) \leq r_T^a(\mathcal{H}_f) \leq \tilde{O}(T^{2/3})$ | Example 2 |

**Related Work** In this paper we study sequential minimax regret for general online regression with logarithmic loss using tools of information theory, in particular universal source coding (lower bounds) [1, 10, 18, 21, 26, 27, 28, 38] and sequential covering (upper bounds).

Most of the existing works in online regression deals with logistic regression. We first mention the work of [13], who studied pointwise regret of logistic regression for the *proper* setting. Unlike *improper* learning, studied in this paper, where feature $\mathbf{x}_t$ at time $t$ is also available to the learner, [13] showed that pointwise regret is $\Theta(T^{1/3})$ for $d = 1$ and $O(\sqrt{T})$ for $d > 1$. Furthermore, [17] demonstrates that regret for logistic regression grows as $O(d \log T/d)$. This was further generalized in [12]. These results were strengthened [29], which also provides matching lower bounds. Precise asymptotics for the fixed design minimax regret were recently presented in [14, 15].

Regret bounds under logarithmic loss for general expert class $\mathcal{H}$ was first investigated by Vovk under the framework of mixable losses [16, 34]. In particular, Vovk showed that for finite class $\mathcal{H}$, the regret growth is $\log |\mathcal{H}|$ via the *aggregating algorithm* (i.e., the Bayesian algorithm that we will discuss below). We refer the reader to [5, Chapter 3.5, 3.6] and the references therein for more results on this topic. Cesa-Bianchi and Lugosi [5] were the first to investigate log-loss under general (infinite) expert class $\mathcal{H}$ [5, Chapter 9.10, 9.11], where they derived a general upper bound using the concept of covering number and a two-stage prediction scheme. In particular, Cesa-Bianchi and Lugosi showed that for Lipschitz parametric classes with values bounded away from $\{0, 1\}$, one can achieve a regret bound of the form $d/2 \log(T/d)$. When the values are close to $\{0, 1\}$, they used a *hard* truncation approach, which gives a sub-optimal bound of the form $3/2d \log(T/d)$ (i.e., this bound is not explicitly shown in [5] but can be derived using their approach). Moreover, the approach of [5] only works for fixed design regret (or *simulatable* in their context). In [23], the authors extended the result of [5, Chapter 9.10] to the sequential case via the machinery of sequential covering that was

established in [22]. However, [23] also used the same *hard* truncation as in [5] resulting in suboptimal upper bounds. In [3], the authors obtained an upper bound similar to the upper bound presented in Theorem 1 via the observation that the $\log$ function is self-concordant. In particular, this allows them to resolve the tight bounds for non-parametric Lipschitz functions that map $[0,1]^s \to [0,1]$. However, their bounds are proved *non-constructively*, i.e., the proof does not provide an algorithm that achieves such bounds. In [4], the authors used a similar idea of smoothing for controlling the unboundedness of log-loss, however, they are assumed that the features $\mathbf{x}^T$ are presented *i.i.d.*. More importantly, the results in [4] only holds for the *average case* regret.

## 2 Problem Formulation and Preliminaries

We denote $\mathcal{X}$ as the input feature space and $\mathcal{H}$ as the concept class, which is a set of functions mapping $\mathcal{X} \to [0,1]$. We use an auxiliary set $\mathcal{W}$ to index $\mathcal{H}$. We say that a function $g$ is *sequential* if it maps $\mathcal{X}^* \to [0,1]$, where $\mathcal{X}^*$ is set of all finite sequences with elements in $\mathcal{X}$. We denote $\mathcal{G}$ as a class of *sequential* functions. If $T$ is a time horizon, then for any $t \in [T]$, we write $\mathbf{x}^t = \{\mathbf{x}_1, \cdots, \mathbf{x}_t\}$ and $y^t = \{y_1, \cdots, y_t\}$. We use standard asymptotic notation $f(t) = O(g(t))$ if there exists a constant $C$ such that $f(t) \leq Cg(t)$ for sufficient large $t \geq 0$, and $f(t) \ll g(t)$ if $\limsup_{t\to\infty} f(t)/g(t) = 0$. We assume the log function $\log(x)$ is the nature logarithm to the base of $e$.

The main objective of this paper is to study the growth of the sequential minimax regret $r_T^a(\mathcal{H})$ for a large class of experts $\mathcal{H}$. We accomplish this goal using two different techniques. For the lower bound, we precisely estimate the fixed design minimax regret $r_T^*(\mathcal{H}|\mathbf{x}^T)$ using the Shtarkov sum [31], discussed next. For the upper bound, we construct a global cover set $\mathcal{G}$ of $\mathcal{H}$ and design a new (truncated) Bayesian algorithm to find precise bounds with constants that are provably optimal.

**Lower Bounds.** We investigate the lower bound of adversarial regret $r_T^a(\mathcal{H})$ by considering its corresponding fixed design minimax regret $r_T^*(\mathcal{H}|\mathbf{x}^T)$ and $r_T^*(\mathcal{H}) = \max_{\mathbf{x}^T} r_T^*(\mathcal{H}|\mathbf{x}^T)$. We are able to do this due to the recent result [37], which we quote next.

**Lemma 1** (Wu et al., 2022). *Let $\mathcal{H}$ be any general hypothesis class and $\ell$ be any loss function. Then*

$$r_T^a(\mathcal{H}) \geq r_T^*(\mathcal{H}),$$

*and the inequality is strict for certain $\mathcal{H}$, and loss function $\ell$.*

We establish precise growth of $r_T^*(\mathcal{H})$ by estimating the Shtarkov sum that was intensively analyzed in information theory [31, 10] and recently applied in online learning [30, 14]. For the logarithmic loss, the Shtarkov sum (conditioned on $\mathbf{x}^T$) is defined as follows [1]

$$S_T(\mathcal{H}|\mathbf{x}^T) \overset{\text{def}}{=} \sum_{y^T \in \{0,1\}^T} \sup_{h \in \mathcal{H}} P_h(y^T \mid \mathbf{x}^T),$$

where $P_h(y^T \mid \mathbf{x}^T) = \prod_{t=1}^T h(\mathbf{x}_t)^{y_t}(1 - h(\mathbf{x}_t))^{1-y_t}$ and we *interpret* $h(\mathbf{x}_t) = P(y_t = 1|\mathbf{x}_t)$. The regret can be expressed in terms of the Shtarkov sum (see [14, Equation (6)] or [5, Theorem 9.1]) as

$$r_T^*(\mathcal{H}) = \sup_{\mathbf{x}^T} \log S_T(\mathcal{H}|\mathbf{x}^T). \tag{5}$$

It is known that the leading term in the Shtarkov sum for parametric classes $\mathcal{H}$ is often independent of $\mathbf{x}^T$ [29, 12, 14, 15]. Therefore, the Shtarkov sum often gives the leading growth of $r_T^*(\mathcal{H}|\mathbf{x}^T)$ independent of $\mathbf{x}^T$, which also suggests the leading growth of the agnostic regret $r_T^a(\mathcal{H})$.

**Upper Bounds.** We now discuss our constructive approach to upper bounds. In the next section, we present our Smooth truncated Bayesian Algorithm (Algorithm 2) that provides a constructive and often achievable upper bound. Here we focus on some, mostly known, preliminaries.

Let $\mathcal{G} \subset [0,1]^{\mathcal{X}^*}$ be any reference class. Let $\mathcal{W}$ be an index set of $\mathcal{G}$ and $\mu$ be an arbitrary finite measure over $\mathcal{W}$. The standard Bayesian predictor with prior $\mu$ is presented in Algorithm 1. Based on this algorithm, we have the following two lemmas that are used to establish most of the upper bounds in this paper. See e.g., [19, Lemma 3] or [5, Chapter 3.3] for proofs.

---

[1] Note that the Starkov sum can be defined for any class of measures, however, here we only use the form for product measures.

**Algorithm 1** Bayesian predictor
___
**Input**: Reference class $\mathcal{G} := \{g_w : w \in \mathcal{W}\}$ with index set $\mathcal{W}$ and prior $\mu$ over $\mathcal{W}$
1: Set $p_w(y^0 \mid \mathbf{x}^0) = 1$ for all $w \in \mathcal{W}$.
2: **for** $t = 1, \cdots, T$ **do**
3:      Receive feature vector $\mathbf{x}_t$
4:      Make prediction using the following equation:
$$\hat{y}_t = \frac{\int_{\mathcal{W}} g_w(\mathbf{x}^t) p_w(y^{t-1} \mid \mathbf{x}^{t-1}) \mathrm{d}\mu}{\int_{\mathcal{W}} p_w(y^{t-1} \mid \mathbf{x}^{t-1}) \mathrm{d}\mu}.$$
5:      Receive label $y_t$
6:      For all $w \in \mathcal{W}$, update: $p_w(y^t \mid \mathbf{x}^t) = e^{-\ell(g_w(\mathbf{x}^t), y_t)} p_w(y^{t-1} \mid \mathbf{x}^{t-1})$.
7: **end for**
___

**Lemma 2.** *Let $\mathcal{G}$ be a collection of functions $g_w : \mathcal{X}^* \to [0, 1], w \in \mathcal{W}$. Let $\hat{y}_t$ be the Bayesian prediction rule as in Step 4 of Algorithm 1 with prior $\mu$. Then, for any $\mathbf{x}^T$ and $y^T$ we have*

$$\sum_{t=1}^T \ell(\hat{y}_t, y_t) \leq -\log \frac{\int_{\mathcal{W}} p_w(y^T \mid \mathbf{x}^T) \mathrm{d}\mu}{\int_{\mathcal{W}} 1 \mathrm{d}\mu},$$

*where $p_w(y^T \mid \mathbf{x}^T) = e^{-\sum_{t=1}^T \ell(g_w(\mathbf{x}^t), y_t)}$ and $\ell$ is the log-loss as in equation (1).*

The following lemma bounds the regret under log-loss of finite classes, which is well known.

**Lemma 3.** *For any finite class of experts $\mathcal{G}$, we have $r_T^a(\mathcal{G}) \leq \log |\mathcal{G}|$.*

## 3 Main Results

We start with a concept of covering set called the *global sequential cover* that was used implicitly in [24, Section 6.1] for the Lipschitz losses and dated back to the ideas in [2].

**Definition 1** (Global sequential covering). *For any $\mathcal{H} \subset [0, 1]^{\mathcal{X}}$, we say that class of sequential functions $\mathcal{G} \subset [0, 1]^{\mathcal{X}^*}$ is a global* sequential $\alpha$-covering *of $\mathcal{H}$ at scale $T$ if for any $\mathbf{x}^T \in \mathcal{X}^T$ and $h \in \mathcal{H}$, there exists $g \in \mathcal{G}$ such that $\forall t \in [T]$,*

$$|h(\mathbf{x}_t) - g(\mathbf{x}^t)| \leq \alpha.$$

*Throughout we assume that $0 \leq \alpha \leq 1$.*

Note that the *global* sequential covering is different from the (local) sequential covering used in [3] (originally from [24]), since our covering function *does not* depend on the underlying trees as in [24] [2]. This is crucial to apply our covering set directly in an algorithmic way (see Algorithm 2). Particularly, it enables us to establish our lower and upper bounds for Lipschitz classes of functions with the optimal constants on the leading term. We further improve these results for Lipschitz class with bounded Hessian. Finally, we study cases when the data dimension $d$ grows faster than $T$ by bounding the covering size through the sequential fat-shattering number. In particular, we prove matching (up to poly $\log T$ factor) upper and lower bounds for the generalized linear functions.

**General Results.** We are now in the position to state our first main general finding.

**Theorem 1.** *If for any $\alpha > 0$ there exists a global sequential $\alpha$-covering set $\mathcal{G}_\alpha$ of $\mathcal{H}$, then*

$$r_T^a(\mathcal{H}) \leq \inf_{0 \leq \alpha \leq 1} \{2\alpha T + \log |\mathcal{G}_\alpha|\}, \tag{6}$$

*and this bound is achievable by Algorithm 2.*

We should point out that Theorem 1 also improves the results of [3] by obtaining better constants in front of both $\alpha T$ and $\log |\mathcal{G}_\alpha|$ (i.e., from $(4, 4)$ to $(2, 1)$). The proof is based on the following key lemma that is established in Appendix A.

___
[2]Note that the covering functions in Definition 1 can be viewed as the experts constructed in [24, Section 6.1]

---
**Algorithm 2** Smooth truncated Bayesian predictor
---
**Input**: Reference class $\mathcal{G}$ with index set $\mathcal{W}$ and prior $\mu$ over $\mathcal{W}$, and truncation parameter $\alpha$

1: Let $p_w(y^0 \mid \mathbf{x}^0) = 1$ for all $w \in \mathcal{W}$
2: **for** $t = 1, \cdots, T$ **do**
3:     Receive feature $\mathbf{x}_t$
4:     For all $w \in \mathcal{W}$, set
$$\tilde{g}_w(\mathbf{x}^t) = \frac{g_w(\mathbf{x}^t) + \alpha}{1 + 2\alpha}$$
5:     Make prediction
$$\hat{y}_t = \frac{\int_{\mathcal{W}} \tilde{g}_w(\mathbf{x}^t) p_w(y^{t-1} \mid \mathbf{x}^{t-1}) \mathrm{d}\mu}{\int_{\mathcal{W}} p_w(y^{t-1} \mid \mathbf{x}^{t-1}) \mathrm{d}\mu}$$
6:     Receive label $y_t$
7:     For all $w \in \mathcal{W}$, update: $p_w(y^t \mid \mathbf{x}^t) = e^{-\ell(\tilde{g}_w(\mathbf{x}^t), y_t)} p_w(y^{t-1} \mid \mathbf{x}^{t-1})$.
8: **end for**
---

**Lemma 4.** *Suppose $\mathcal{H}$ has a global sequential $\alpha$-covering set $\mathcal{G}$ for some $\alpha \in [0, 1]$. Then, there exists a truncated set $\tilde{\mathcal{G}}$ of $\mathcal{G}$ with $|\tilde{\mathcal{G}}| = |\mathcal{G}|$ such that for all $\mathbf{x}^T, y^T$ and $h \in \mathcal{H}$ there exists a $\tilde{g} \in \tilde{\mathcal{G}}$ satisfying*

$$\frac{p_h(y^T \mid \mathbf{x}^T)}{p_{\tilde{g}}(y^T \mid \mathbf{x}^T)} \le (1 + 2\alpha)^T, \tag{7}$$

*where*

$$p_h(y^T \mid \mathbf{x}^T) = \prod_{t=1}^{T} h(\mathbf{x}_t)^{y_t} (1 - h(\mathbf{x}_t))^{1-y_t} \quad \text{and} \quad p_{\tilde{g}}(y^T \mid \mathbf{x}^T) = \prod_{t=1}^{T} \tilde{g}(\mathbf{x}^t)^{y_t} (1 - \tilde{g}(\mathbf{x}^t))^{1-y_t}.$$

*Proof of Theorem 1.* We show that for any $0 \le \alpha \le 1$ if an $\alpha$-covering set $\mathcal{G}_\alpha$ exists, then one can achieve the claimed bound for such an $\alpha$. To do so, we run the Smooth truncated Bayesian Algorithm (Algorithm 2) on $\mathcal{G}_\alpha$ with uniform prior and truncation parameter $\alpha$. We denote by $\tilde{\mathcal{G}}_\alpha$ the truncated class of $\mathcal{G}_\alpha$ as in Lemma 4 (same as the step 4 of Algorithm 2). We now fix $\mathbf{x}^T, y^T$. By Lemma 3 (with $\mathcal{G}$ being $\tilde{\mathcal{G}}_\alpha$), we have

$$\sum_{t=1}^{T} \ell(\hat{y}_t, y_t) \le \inf_{\tilde{g} \in \tilde{\mathcal{G}}_\alpha} \sum_{t=1}^{T} \ell(\tilde{g}(\mathbf{x}^t), y_t) + \log |\tilde{\mathcal{G}}_\alpha| = \inf_{\tilde{g} \in \tilde{\mathcal{G}}_\alpha} \sum_{t=1}^{T} \ell(\tilde{g}(\mathbf{x}^t), y_t) + \log |\mathcal{G}_\alpha|,$$

the last equality follows from $|\mathcal{G}_\alpha| = |\tilde{\mathcal{G}}_\alpha|$. Since $\sum_{t=1}^{T} \ell(f(\mathbf{x}^t), y_t) = -\log p_f(y^T \mid \mathbf{x}^T)$ for any function $f$, then by Lemma 4 we conclude that

$$\inf_{h \in \mathcal{H}} \sum_{t=1}^{T} \ell(h(\mathbf{x}_t), y_t) \ge \inf_{\tilde{g} \in \tilde{\mathcal{G}}_\alpha} \sum_{t=1}^{T} \ell(\tilde{g}(\mathbf{x}^t), y_t) - T \log (1 + 2\alpha).$$

The result follows by combining the inequalities and noticing that $\log(1+x) \le x$ for all $x \ge -1$. $\square$

We further note that for any constant $c_1, c_2$ for which the bound $r_T^a(\mathcal{H}) \le c_1 \alpha T + c_2 \log |\mathcal{G}_\alpha|$ holds universally we must have $c_1 \ge 2$ and $c_2 \ge 1$. Therefore, our bounds are optimal with respect to the constants[3]. To see this, we let $\mathcal{X} = [T]$ and define $g$ to be the function that maps every $t \in [T]$ to $\frac{1}{2}$. Let $\mathcal{H}$ be the class of functions that maps to $[1/2 - \alpha, 1/2 + \alpha]$. Clearly, $\mathcal{H}$ is $\alpha$-covered by $g$. By noting that the maximum probability is $(1/2 + \alpha)^T = (1 + 2\alpha)^T (1/2)^T$, we compute the Shtarkov sum (5) to get:

$$r_T^a(\mathcal{H}) \ge r_T^*(\mathcal{H}) \ge \log(1 + 2\alpha)^T \sim 2\alpha T,$$

where $\sim$ holds when $\alpha$ is sufficiently small. This implies that we must have $c_1 \ge 2$. The fact that $c_2 \ge 1$ is due to the fact that mixability constant of log-loss is 1, which also follows from Theorem 3 below.

---

[3]Note that the optimally only shows that the constants in the form $2\alpha T + \log |\mathcal{G}_\alpha|$ cannot be improved. However, it is quite possible that one can obtain better bounds with a different form.

**Lipschitz Parametric Class.** We now consider a Lipschitz parametric function class. Given a function $f : \mathcal{W} \times \mathcal{X} \to [0, 1]$, define the following class

$$\mathcal{H}_f = \{f(\mathbf{w}, \cdot) \in [0, 1]^{\mathcal{X}} \; : \; \mathbf{w} \in \mathcal{W}\},$$

where $\mathbf{w} \in \mathcal{W}$ is often a vector in $\mathbb{R}^d$.

We will assume that $f(\mathbf{w}, \mathbf{x})$ is $L$-Lipschitz on $\mathbf{w}$ for every $\mathbf{x}$, where $L \in \mathbb{R}^+$. More formally, $\forall \mathbf{w}_1, \mathbf{w}_2 \in \mathcal{W}$ and $\mathbf{x} \in \mathcal{X}$ we have

$$|f(\mathbf{w}_1, \mathbf{x}) - f(\mathbf{w}_2, \mathbf{x})| \leq L||\mathbf{w}_1 - \mathbf{w}_2||,$$

where $|| \cdot ||$ is some norm on $\mathcal{W}$. For example, if we take $\mathcal{W} \subset \mathbb{R}^d$ then the norm can be $\ell_1$, $\ell_2$ or $\ell_\infty$ norm. For any specific norm $|| \cdot ||$, we write $\mathcal{B}(R)$ for the ball under such norm with radius $R$ in $\mathcal{W}$. In particular, we denote by $\mathcal{B}_s^d(R)$ the ball in $\mathbb{R}^d$ of radius $R$ under $\ell_s$ norm centered at the origin.

**Theorem 2.** *Let $f : \mathcal{B}_s^d(R) \times \mathbb{R}^d \to [0, 1]$ be a $L$-Lipschitz function under $\ell_s$ norm. Then*

$$r_T^a(\mathcal{H}_f) \leq \min\left\{d\log\left(\frac{2RLT}{d} + 1\right) + 2d, T\right\}. \tag{8}$$

*Proof.* By $L$-Lipschitz condition, to find an $\alpha$-covering in the sense of Definition 1, we only need to find a covering of $\mathcal{B}_s^d(R)$ with radius $\alpha/L$. By standard result (see e.g. Lemma 5.7 and Example 5.8 of [35]) we know that the covering size is upper bounded by

$$\left(\frac{2RL}{\alpha} + 1\right)^d.$$

By Theorem 1, we find

$$r_T^a(\mathcal{H}_f) \leq \inf_{0 < \alpha < 1}\left\{2\alpha T + d\log\left(\frac{2RL}{\alpha} + 1\right)\right\}.$$

Taking $\alpha = d/T$, we conclude

$$r_T^a(\mathcal{H}_f) \leq d\log\left(\frac{2RLT}{d} + 1\right) + 2d.$$

This completes the proof for $T \geq d$. The upper bound $T$ is achieved by predicting $\frac{1}{2}$ every time. $\square$

**Example 1.** For logistic function $f(\mathbf{w}, \mathbf{x}) = (1 + e^{-\langle \mathbf{w}, \mathbf{x} \rangle})^{-1}$, and $\mathbf{w} \in \mathcal{B}_2^d(R)$ with $\mathbf{x} \in \mathcal{B}_2^d(1)$ our result recovers those of [12], but with a better leading constant (the bound in [12] has a constant 5). Note that, the result in [3] also provides a sub-optimal constant $c \sim 4$. Moreover, our bounds have a logarithmic dependency on Lipschitz constant $L$.

The question arises whether the factor in front of $\log T$ can be improved to $d/2$ instead of $d$ as discussed in some recent papers [29, 14, 15]. In Theorem 3 below, we show that, in general, it cannot unless we further strengthen our assumption (see Theorem 4). For the ease of presentation, we only consider the parameters restricted to $\ell_2$ norm. The proof can be found in Appendix B.

**Theorem 3.** *For any $d, T, R, L$ such that $T \gg d\log(RLT)$, there exists $L$-Lipschitz function $f : \mathcal{B}_2^d(R) \times \mathbb{R}^d \to [0, 1]$ such that*

$$r_T^a(\mathcal{H}_f) \geq d\log\left(\frac{RLT}{d}\right) - d\log 64 - d\log\log(RLT). \tag{9}$$

**Lipschitz Class with Bounded Hessian.** As we have demonstrated in Theorem 3 the leading constant 1 of the regret for Lipschitz parametric classes can not be improved in general. We now show that for some special function $f$ one can improve the constant to $\frac{1}{2}$, as already noticed in [29, 14, 15]. For any function $f : \mathbb{R}^d \times \mathbb{R}^d \to [0, 1]$, we say the Hessian of $\log f$ is uniformly bounded on $\mathcal{X} \subset \mathbb{R}^d$, if there exists constant $C$ such that for any $\mathbf{w} \in \mathbb{R}^d$ and $\mathbf{x} \in \mathcal{X}$ and $y \in \{0, 1\}$ we have

$$\sup_{||\mathbf{u}||_2 \leq 1} |\mathbf{u}^T \nabla_{\mathbf{w}}^2 \log f(\mathbf{w}, \mathbf{x})^y (1 - f(\mathbf{w}, \mathbf{x}))^{1-y} \mathbf{u}| \leq C,$$

where $\nabla_{\mathbf{w}}^2$ is the Hessian at $\mathbf{w}$. The proof of the next theorem can be found in Appendix C.

**Theorem 4.** *Let $f : \mathbb{R}^d \times \mathbb{R}^d \to [0,1]$ be a function such that the Hessian of $\log f$ is uniformly bounded by $C$ on $\mathcal{X}$. Let*

$$\mathcal{H}_f = \{f(\mathbf{w}, \mathbf{x}) : \mathbf{w} \in \mathcal{W}, \mathbf{x} \in \mathcal{X}\}$$

*be such a class of $f$ restricted to some compact set $\mathcal{W} \subset \mathbb{R}^d$. Then*

$$r_T^a(\mathcal{H}_f) \leq \log \frac{Vol(\mathcal{W}^*)}{Vol(\mathcal{B}_2^d(\sqrt{d/CT}))} + d/2 + \log 2. \tag{10}$$

*where $\mathcal{W}^* = \{\mathbf{w} + \boldsymbol{u} \mid \mathbf{w} \in \mathcal{W}, \boldsymbol{u} \in \mathcal{B}_2^d(\sqrt{d/CT})\}$, $Vol(\cdot)$ is volume under Lebesgue measure. In particular, for $\mathcal{W} = \mathcal{B}_2^d(R)$, we have*

$$r_T^a(\mathcal{H}_f) \leq \frac{d}{2} \log \left( \frac{2CR^2 T}{d} + 2 \right) + d/2 + \log 2.$$

Note that, Theorem 4 subsumes the results of [17, 29][4], where the authors considered functions of form $f(\langle \mathbf{w}, \mathbf{x} \rangle)$ and requires that the second derivative of $\log f$ is bounded, see also [5, Chapter 11.10]. However, the KL-divergence-based argument of [17] can not be used directly in the setup of Theorem 4 since we *do not* assume the function $f$ has a linear structure. Our main proof technique of Theorem 4 is a direct application of Lemma 2 and an estimation of the integrals via Taylor expansion; see Appendix C for more details on the proof.

Finally, we complete this part with the following lower bound for generalized linear functions under unit $\ell_s$ balls. See Appendix D for proof.

**Theorem 5.** *Let $f : \mathbb{R} \to [0,1]$ be an arbitrary function such that there exists $c_1, c_2 \in (0,1)$ and for all $r > 0$ we have $[c_1 - c_2 d^{-r}, c_1 + c_2 d^{-r}] \subset f([-d^{-r}, d^{-r}])$ for sufficiently large $d$. Let*

$$\mathcal{H}_f = \{f(\langle \mathbf{w}, \mathbf{x} \rangle) : \mathbf{w} \in \mathcal{B}_s^d(1), \mathbf{x} \in \mathcal{B}_s^d(1)\}$$

*where $s > 0$. Then*

$$r_T^a(\mathcal{H}_f) \geq \frac{d}{2} \log \left( \frac{T}{d^{(s+2)/s}} \right) - O(d) \tag{11}$$

*where $O$ hides some absolute constant that is independent of $d, T$.*

Note that for the logistic function $f(x) = (1 + e^{-x})^{-1}$ Theorem 5 holds with $c_1 = \frac{1}{2}$ and $c_2 = \frac{1}{5}$. Therefore,
**1**. If $s = 1$, then

$$r_T^a(\mathcal{H}_f) \geq \frac{d}{2} \log \left( \frac{T}{d^3} \right) - O(d).$$

**2**. If $s = 2$, then

$$r_T^a(\mathcal{H}_f) \geq \frac{d}{2} \log \left( \frac{T}{d^2} \right) - O(d).$$

**3**. If $s = \infty$, then

$$r_T^a(\mathcal{H}_f) \geq \frac{d}{2} \log \left( \frac{T}{d} \right) - O(d).$$

This recovers all the lower bounds from [29]. We note that a simple sufficient condition for Theorem 5 to hold is to require $f'(0) \neq 0$ if $f(x)$ is differentiable.

**Large Growth.** We now present some results for large $d$ growing even faster than $T$. We will show that the size of *global* sequential covering (Definition 1) of a class $\mathcal{H}$ can be bounded by the sequential fat-shattering number of $\mathcal{H}$ in a similar fashion as in [24]. We first introduce the notion of sequential fat-shattering number as in [24].

We denote $\{0,1\}_*^d$ to be the set of all binary sequences of length less than or equal to $d$. A binary tree of depth $d$ with labels in $\mathcal{X}$ is defined to be a map $\tau : \{0,1\}_*^d \to \mathcal{X}$. For any function class $\mathcal{H} \subset [0,1]^{\mathcal{X}}$, we say $\mathcal{H}$ $\alpha$-fat shatters tree $\tau$ if there exists $[0,1]$-value tree $\mathbf{s} : \{0,1\}_*^d \to [0,1]$ such that for any binary sequence $\epsilon_1^d \in \{0,1\}_*^d$ there exist $h \in \mathcal{H}$ such that for all $t \in [d]$:
1. If $\epsilon_t = 0$, then $h(\tau(\epsilon_1^{t-1})) \leq \mathbf{s}(\epsilon_1^{t-1}) - \alpha$;
2. If $\epsilon_t = 1$, then $h(\tau(\epsilon_1^{t-1})) \geq \mathbf{s}(\epsilon_1^{t-1}) + \alpha$.

---

[4]To get the upper bounds in [29] one only needs to estimate the volume of $\ell_s$ balls, which is well known [36].

**Definition 2.** *The sequential $\alpha$-fat shattering number of $\mathcal{H}$ is defined to be the maximum number $d(\alpha)$ such that $\mathcal{H}$ $\alpha$-fat shatters a tree $\tau$ of depth $d := d(\alpha)$.*

In the lemma below, we present an upper bound for the cardinality of the global covering with algorithmically constructed cover set $\mathcal{G}_\alpha$, see e.g., [24, Section 6.1]. We provide a proof in Appendix E for completeness.

**Lemma 5.** *Let $\mathcal{H} \subset [0,1]^{\mathcal{X}}$ be any class and $d(\alpha)$ be the sequential $\alpha$-fat shattering number of $\mathcal{H}$. Then there exists a global sequential $\alpha$-covering set $\mathcal{G}_\alpha$ of $\mathcal{H}$ as in Definition 1 such that*

$$|\mathcal{G}_\alpha| \leq \sum_{t=0}^{d(\alpha/3)} \binom{T}{t} \left\lceil \frac{3}{2\alpha} \right\rceil^t \leq \left\lceil \frac{3T}{2\alpha} \right\rceil^{d(\alpha/3)+1}. \tag{12}$$

**Example 2.** By [24] we know that the sequential $\alpha$-fat shattering number of linear functions $f(\mathbf{w}, \mathbf{x}) = |\langle \mathbf{w}, \mathbf{x} \rangle|$ with $\mathbf{w}, \mathbf{x} \in \mathcal{B}_2^d(1)$ is of order $\tilde{O}(\alpha^{-2})$ where in $\tilde{O}$ we hide a polylog factor. Lemma 5 implies that the global sequential $\alpha$-covering number is upper bounded by

$$\left\lceil \frac{(3T)}{(2\alpha)} \right\rceil^{d(\alpha/3)+1}.$$

By Theorem 1, we have

$$r_T^a(\mathcal{H}_f) \leq \inf_{0 < \alpha < 1} \left\{ 2\alpha T + \tilde{O}\left(\frac{1}{\alpha^2}\right) \right\} \leq \tilde{O}(T^{2/3}),$$

by taking $\alpha = T^{-1/3}$. This bound is *independent* of the data dimension $d$.

**Remark 1.** *Observe that for any class $\mathcal{H}$ with sequential fat-shattering number of order $\alpha^{-s}$ one can achieve a regret upper bound of order $\tilde{O}(T^{s/s+1})$ by Theorem 1. We refer to [24, 25] for the estimations of sequential fat-shattering number of a variety of classes.*

Finally, we present the following general lower bound. See Appendix F for proof.

**Theorem 6.** *For any $s \geq 1$, we define*

$$\mathcal{D}_s = \left\{ \mathbf{p} \in [0,1]^T : \sum_{t=1}^T p_t^s \leq 1 \right\}.$$

*We can view the vectors in $\mathcal{D}_s$ as functions mapping $[T] \to [0,1]$. Then*

$$r_T^a(\mathcal{D}_s) \geq r_T^*(\mathcal{D}_s) \geq \Omega(T^{s/s+1}). \tag{13}$$

To see why Theorem 6 implies a lower bound for $f(\mathbf{w}, \mathbf{x}) = |\langle \mathbf{w}, \mathbf{x} \rangle|$ with $d \geq T$, as in Example 2, we take $\mathbf{w}, \mathbf{x} \in \mathcal{B}_2^T(1)$ (i.e., with $d = T$) and define $\mathbf{x}_t = \mathbf{e}_t$ with $\mathbf{e}_t$ being the standard base of $\mathbb{R}^T$ that takes value 1 at position $t$ and zeros otherwise. Note that the functions of $\mathcal{H}_f$ with $f(\mathbf{w}, \mathbf{x}) = |\langle \mathbf{w}, \mathbf{x} \rangle|$ restricted on $\mathbf{x}^T$ is exactly $\mathcal{D}_2$. Then

$$r_T^a(\mathcal{H}_f) \geq r_T^*(\mathcal{H}_f) \geq r_T^*(\mathcal{D}_2) \geq \Omega(T^{2/3})$$

and this is a matching lower bound of Example 2. Note that, it is proved in [23] that for function $f(\mathbf{w}, \mathbf{x}) = \frac{\langle \mathbf{w}, \mathbf{x} \rangle + 1}{2}$, one can achieve the regret of form $\tilde{O}(\sqrt{T})$[5]. Example 2 implies that the generalized linear functions of form $f(\langle \mathbf{w}, \mathbf{x} \rangle)$ can have different regrets with polynomial gap even with a simple shift on the value (though they have the same covering number). It is therefore an interesting open problem to investigate a tighter complexity measure (instead of a covering number as in Definition 1) that captures this phenomenon.

## 4 Conclusion

In this paper, we presented best known lower and upper bounds on sequential online regret for a large class of experts. We accomplish this by designing a new smooth truncated Bayesian algorithm, together with the concept of global sequential covering, that achieves these upper bounds. For the lower bound, we use a novel information-theoretic approach based on the Shtarkov sum. We expect that these techniques can be generalized to a broader set of problems, e.g., when the features $\mathbf{x}^T$ is present stochastically. We leave these to the future investigations.

---

[5]A $\tilde{\Omega}(\sqrt{T})$ lower bound for $d \geq \sqrt{T}$ can be derived from Theorem 5, recovering [23, Lemma 8].

## Acknowledgments

This work was partially supported by the NSF Center for Science of Information (CSoI) Grant CCF-0939370, by NSF Grants CCF-2006440, CCF-2007238, and CCF- 2211423, and in addition by Google Research Grant and by Rolls Royce.

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
