# A  Proof of Lemma 4

We construct the set $\tilde{\mathcal{G}}$ as in Algorithm 2. For any $g \in \mathcal{G}$ we define a smooth truncated function $\tilde{g}$ such that for any $\mathbf{x}^t \in \mathcal{X}^*$

$$\tilde{g}(\mathbf{x}^t) = \frac{g(\mathbf{x}^t) + \alpha}{1 + 2\alpha}.$$

We introduce the following short-hand notation, for any function $f$ we define

$$f(y_t) = f(\mathbf{x}^t)^{y_t}(1 - f(\mathbf{x}^t))^{1-y_t}.$$

For any $\mathbf{x}^T$, $y^T$ and $h \in \mathcal{H}$, let $g \in \mathcal{G}$ be a $\alpha$-covering of $h$ and $\tilde{g}$ be the truncated function as defined above. For any $t$, we consider two cases.

**Case 1**: If $y_t = 1$, we have:

$$\frac{h(y_t)}{\tilde{g}(y_t)} = \frac{h(\mathbf{x}^t)}{\tilde{g}(\mathbf{x}^t)}, \text{ since } y_t = 1 \tag{14}$$

$$\leq \frac{g(\mathbf{x}^t) + \alpha}{\tilde{g}(\mathbf{x}^t)}, \ g \text{ is } \alpha \text{ cover of } h \tag{15}$$

$$= \frac{g(\mathbf{x}^t) + \alpha}{(g(\mathbf{x}^t) + \alpha)/(1 + 2\alpha)}, \text{ definition of } \tilde{g} \tag{16}$$

$$= 1 + 2\alpha \tag{17}$$

**Case 2**: If $y_t = 0$, we have

$$\frac{h(y_t)}{\tilde{g}(y_t)} = \frac{1 - h(\mathbf{x}^t)}{1 - \tilde{g}(\mathbf{x}^t)} \tag{18}$$

$$\leq \frac{1 - g(\mathbf{x}^t) + \alpha}{1 - \tilde{g}(\mathbf{x}^t)}, \ g \text{ is } \alpha \text{ cover of } h \tag{19}$$

$$= \frac{1 - g(\mathbf{x}^t) + \alpha}{1 - (g(\mathbf{x}^t) + \alpha)/(1 + 2\alpha)}, \text{ definition of } \tilde{g} \tag{20}$$

$$= \frac{1 - g(\mathbf{x}^t) + \alpha}{(1 - g(\mathbf{x}^t) + \alpha)/(1 + 2\alpha)} \tag{21}$$

$$= 1 + 2\alpha, \tag{22}$$

Now, combining the two cases, we have

$$\frac{p_h(y^T \mid \mathbf{x}^T)}{p_{\tilde{g}}(y^T \mid \mathbf{x}^T)} = \prod_{t=1}^{T} \frac{h(y_t)}{\tilde{g}(y_t)} \tag{23}$$

$$\leq (1 + 2\alpha)^T. \tag{24}$$

This completes the proof of Lemma 4.

# B  Proof of Theorem 3

We need the following two lemmas, where the proofs are straightforward.

**Lemma 6.** *Let $\mathcal{P}$ be a finite class of distributions over the same domain $\mathcal{X}$. Denote*

$$S = \sum_{x \in \mathcal{X}} \max_{p \in \mathcal{P}} p(x)$$

*be the Shtarkov sum. Then for any estimation rule $\Phi : \mathcal{X} \to \mathcal{P}$ we have*

$$S \geq |\mathcal{P}| \cdot \left( 1 - \max_{p \in \mathcal{P}} \mathrm{Pr}_{x \sim p}\left[ \Phi(x) \neq p \right] \right)$$

**Lemma 7.** *For any $M$ and $T \gg \log M$, there exist $M$ vectors $v_1, v_2, \cdots, v_M \in \{0,1\}^T$ such that for any $i \neq j \in [M]$ we have*

$$\sum_{t=1}^{T} 1\{v_i[t] \neq v_j[t]\} \geq T/4.$$

Now we are in the position to prove Theorem 3. Let $\mathbf{x}_1, \cdots, \mathbf{x}_T \in \mathbb{R}^d$ be any distinct points. We will construct a $L$-Lipschitz function $f(\mathbf{w}, \mathbf{x})$ such that the regret restricted only on $\mathbf{x}^T$ is large. To do so, we consider a maximum packing $M$ of the parameter space $\mathcal{B}_2^d(R)$ of radius $\alpha/L > 0$ (where $\alpha$ is to be determined latter). Standard volume argument (see Chapter 5 of [35]) yields that

$$|M| \geq \left(\frac{LR}{2\alpha}\right)^d.$$

Now, we will define a $L$-Lipschitz functions $f(\mathbf{w}, \mathbf{x})$ only on $\mathbf{w} \in M$ and $\mathbf{x} \in \{\mathbf{x}_1, \cdots, \mathbf{x}_T\}$. By Lemma 7 (assume for now the conditions are satisfied), we can find $|M|$ binary vectors $V \subset \{0,1\}^T$ such that any pair of the vectors has Hamming distance lower bounded by $T/4$. For each of the vector $v \in V$, we define a vector $u \in [0,1]^T$ in the following way, for all $t \in [T]$

1. If $v[t] = 0$ then set $u[t] = 0$;
2. If $v[t] = 1$ then set $u[t] = \alpha$.

Denote by $U$ be the set of all such vectors $u$. Note that $|U| = |M|$. For any $\mathbf{w} \in M$, we can associate a unique $u \in U$ such that for all $t \in [T]$

$$f(\mathbf{w}, \mathbf{x}_t) = u[t].$$

We now show that $f$ is indeed $L$-Lipschitz restricted on $M$ for all $\mathbf{x}_t \in \{\mathbf{x}_1, \cdots, \mathbf{x}_T\}$. This is because for any $\mathbf{w}_1 \neq \mathbf{w}_2 \in M$ we have $|f(\mathbf{w}_1, \mathbf{x}_t) - f(\mathbf{w}_2, \mathbf{x}_t)| \leq \alpha$ by definition of $U$ and $||\mathbf{w}_1 - \mathbf{w}_2||_2 \geq \alpha/L$ since $M$ is a packing.

We now view the vectors in $u \in U$ as a product of Bernoulli distributions with each coordinate $t$ independently sampled from $\text{Bern}(u[t])$. We show that the sources in $U$ are identifiable. To see this, we note that for any distinct pairs $u_1, u_2 \in U$, there exist a set $I \in [T]$ such that $u_1$ and $u_2$ differ on $I$ and $|I| \geq T/4$. This further implies that there exist a set $J \subset I$ with $|J| \geq T/8$ such that $u_1$ takes all 0 on $J$ and $u_2$ takes all $\alpha$ on $J$ (or vice versa). We can then distinguish $u_1, u_2$ by checking if the samples on $J$ are all 0s or not. The probability of making error is upper bounded by

$$(1 - \alpha)^{T/8} \leq e^{-\alpha T/8}.$$

Since there are only $|M|^2$ such pairs, we have the probability of wrongly identifying the source upper bounded by

$$|M|^2 e^{-\alpha T/8}.$$

Taking $\alpha = \frac{16 d \log(RLT)}{T}$, the error probability is upper bounded by

$$\left(\frac{RLT}{32 d \log(RLT)}\right)^{2d} e^{-2d \log(RLT)} \leq \left(\frac{1}{32 d \log(RLT)}\right)^{2d} \leq \frac{1}{2},$$

for sufficient large $d, T$, where we have use the fact that $|M| \leq (\frac{RLT}{32 d \log(RLT)})^d$. Note that we only showed a lower bound on $|M|$ before, but this is not a problem since we can always remove some points from $M$ to make the upper bound holds as well.

By Lemma 6, we know that the Shtarkov sum of sources in $U$ is lower bounded by $|M|/2$. Therefore, we have

$$r_T^a(\mathcal{H}_f) \geq r_T^*(\mathcal{H}_f) \geq \log(|M|/2) \geq d \log(RLT/d) - d \log 64 - d \log \log(RLT).$$

Now, we have to extend the function to the whole set $\mathcal{B}_2^d(R)$ and keep the $L$-Lipschitz property. This follows from a classical result in real analysis (see [20, Theorem 1]) by defining for all $\mathbf{w} \in \mathcal{B}_2^d(R)$ and $\mathbf{x}_t \in \{\mathbf{x}_1, \cdots, \mathbf{x}_T\}$

$$f(\mathbf{w}, \mathbf{x}_t) = \sup_{\mathbf{w}' \in M} \{f(\mathbf{w}', \mathbf{x}_t) - L||\mathbf{w} - \mathbf{w}'||_2\}.$$

For the $\mathbf{x} \notin \{\mathbf{x}_1, \cdots, \mathbf{x}_T\}$, we can simply let $f(\mathbf{w}, \mathbf{x}) = 0$ for all $\mathbf{w}$.

Finally, we need to check that the condition of Lemma 7 holds for our choice of $\alpha$, this is satisfied by our assumption $T \gg d \log(RLT)$.

## C  Proof of Theorem 4

To make the proof more transparent, we only prove the case for $\mathcal{W} = \mathcal{B}_2^d(R)$ since the proof for other compact $\mathcal{W}$ follows similar path. Note that, for $\mathcal{W} = \mathcal{B}_2^d(R)$, we have $\mathcal{W}^* = \mathcal{B}_2^d(R + \sqrt{d/CT})$.

The proof resembles that of [12] but running the Bayesian predictor (Algorithm 1) over $\mathcal{W}^*$ instead of $\mathcal{W}$ with $\mathcal{G}$ being $\mathcal{H}_f$ and $\mu$ being Lebesgue measure. Let $\mathbf{x}^T$, $y^T$ and $\hat{y}^T$ be the feature, label and predictions of the Bayesian predictor respectively. By Lemma 2

$$\sum_{t=1}^{T} \ell(\hat{y}_t, y_t) \leq -\log \frac{\int_{\mathcal{B}_2^d(R+\sqrt{d/CT})} p_{\mathbf{w}}(y^T \mid \mathbf{x}^T) \mathrm{d}\mu}{\int_{\mathcal{B}_2^d(R+\sqrt{d/CT})} 1 \mathrm{d}\mu}, \tag{25}$$

where $\mu$ is the Lebesgue measure and

$$p_{\mathbf{w}}(y^T \mid \mathbf{x}^T) = \prod_{t=1}^{T} f(\mathbf{w}, \mathbf{x}_t)^{y_t} (1 - f(\mathbf{w}, \mathbf{x}_t))^{1-y_t}.$$

We now write $h_t(\mathbf{w}) \stackrel{\text{def}}{=} \log f(\mathbf{w}, \mathbf{x}_t)^{y_t} (1 - f(\mathbf{w}, \mathbf{x}_t))^{1-y_t}$ to simplify notation. It is easy to see that $\ell(f(\mathbf{w}, \mathbf{x}_t), y_t) = -h_t(\mathbf{w})$.

Let $\mathbf{w}^*$ be the point in $\mathcal{B}_2^d(R)$ that maximizes

$$h(\mathbf{w}) \stackrel{\text{def}}{=} \sum_{t=1}^{T} h_t(\mathbf{w}).$$

Let $\mathbf{u} = \nabla h(\mathbf{w}^*)$ be the gradient of $h$ at $\mathbf{w}^*$. By Taylor theorem, we have for any $\mathbf{w} \in \mathcal{B}_2^d(R + \sqrt{d/CT})$

$$h(\mathbf{w}) = h(\mathbf{w}^*) + \mathbf{u}^T(\mathbf{w} - \mathbf{w}^*) + \frac{1}{2}(\mathbf{w} - \mathbf{w}^*)^\tau \nabla_{\mathbf{w}'}^2 h(\mathbf{w}')(\mathbf{w} - \mathbf{w}^*),$$

where $\mathbf{w}'$ is a convex combination of $\mathbf{w}$ and $\mathbf{w}^*$ and $\mathbf{u}^\tau$ is the transpose of $\mathbf{u}$.

Now, the key observation is that for any point $\mathbf{w}$ such that $\mathbf{u}^\tau(\mathbf{w} - \mathbf{w}^*) \geq 0$ we have

$$h(\mathbf{w}) \geq h(\mathbf{w}^*) + \frac{1}{2}(\mathbf{w} - \mathbf{w}^*)^\tau \nabla_{\mathbf{w}'}^2 h(\mathbf{w}')(\mathbf{w} - \mathbf{w}^*) \geq h(\mathbf{w}^*) - \frac{1}{2} CT \|\mathbf{w} - \mathbf{w}^*\|_2^2, \tag{26}$$

where the last inequality follows from our assumption about the bounded Hessian of $\log f$. Let $B$ be the half ball of radius $\sqrt{d/CT}$ centered at $\mathbf{w}^*$ such that for all $\mathbf{w} \in B$ we have $\mathbf{u}^T(\mathbf{w} - \mathbf{w}^*) \geq 0$. By (26), for all $\mathbf{w} \in B$

$$h(\mathbf{w}) \geq h(\mathbf{w}^*) - \frac{1}{2} CT(\sqrt{d/CT})^2 = h(\mathbf{w}^*) - d/2. \tag{27}$$

Note that $B \subset \mathcal{B}_2^d(R + \sqrt{d/CT})$. Then using above observations we arrive at

$$\sum_{t=1}^{T} \ell(\hat{y}_t, y_t) \leq -\log \frac{\int_{\mathcal{B}_2^d(R+\sqrt{d/CT})} p_{\mathbf{w}}(y^T \mid \mathbf{x}^T) d\mu}{\int_{\mathcal{B}_2^d(R+\sqrt{d/CT})} 1 d\mu} \tag{28}$$

$$\leq -\log \frac{\int_B p_{\mathbf{w}}(y^T \mid \mathbf{x}^T) d\mu}{\int_{\mathcal{B}_2^d(R+\sqrt{d/CT})} 1 d\mu} \tag{29}$$

$$\leq -\log \frac{e^{-d/2} \int_B p_{\mathbf{w}^*}(y^T \mid \mathbf{x}^T) d\mu}{\int_{\mathcal{B}_2^d(R+\sqrt{d/CT})} 1 d\mu} \tag{30}$$

$$= -\log p_{\mathbf{w}^*}(y^T \mid \mathbf{x}^T) + d/2 - \log \frac{\mathrm{Vol}(B)}{\mathrm{Vol}(\mathcal{B}_2^d(R + \sqrt{d/CT}))} \tag{31}$$

$$= -\log p_{\mathbf{w}^*}(y^T \mid \mathbf{x}^T) + d/2 - \log \frac{\frac{1}{2}\sqrt{\frac{d}{CT}}^d}{(R + \sqrt{d/CT})^d} \tag{32}$$

$$\leq -\log p_{\mathbf{w}^*}(y^T \mid \mathbf{x}^T) + d/2 + \frac{d}{2}\log\left(\frac{2CR^2T}{d} + 2\right) + \log 2 \tag{33}$$

$$= \sum_{t=1}^{T} \ell(f(\mathbf{w}^*, \mathbf{x}_t), y_t) + \frac{d}{2}\log\left(\frac{2CR^2T}{d} + 2\right) + d/2 + \log 2. \tag{34}$$

This completes the proof of Theorem 4.

**Remark 2.** *When compared to the technique in [40], Theorem 4 does not assume that the gradient critical point of the loss is zero (e.g., the minimum may occur on the boundary). This is why we need to restrict to the half ball $B$ in order to discard the linear term of Taylor expansion in Equation (27). Moreover, in the proof we work directly on the continuous space instead of a discretized cover, giving an efficient algorithm provided the posterior is efficiently samplable (by e.g., assuming some log-concavity of $f$ as in [12]).*

# D  Proof of Theorem 5

We start with the following technical lemma.[6]

**Lemma 8.** *The following inequality holds, for $r > 0$:*

$$\sum_{\mathbf{y} \in \{0,1\}^{T/d}} \sup_{w \in [c_1 - c_2 d^{-r}, c_1 + c_2 d^{-r}]} P(\mathbf{y} \mid w) \geq \Omega(\sqrt{T/d^{2r+1}}), \tag{35}$$

*where $P(\mathbf{y} \mid w) = w^k(1-w)^{T/d-k}$ with $k$ being the number of $1$s in $\mathbf{y}$.*

*Proof.* By Stirling approximation, for all $k \in [T/d]$, there exists a constant $C \in \mathbb{R}^+$ such that

$$B(k, T/d) \overset{\text{def}}{=} \binom{T/d}{k}\left(\frac{k}{T/d}\right)^k \left(1 - \frac{k}{T/d}\right)^{T/d-k}$$

$$\geq C\sqrt{\frac{T/d}{k(T/d - k)}}.$$

Since $P(\mathbf{y} \mid w)$ achieves maximum at $w = k * d/T$, we have

$$\sum_{\mathbf{y} \in \{0,1\}^{T/d}} \sup_{w \in [c_1 - c_2 d^{-r}, c_1 + c_2 d^{-r}]} p(\mathbf{y} \mid w) \geq \sum_{k=c_1 T/d - c_2 T/d^{r+1}}^{c_1 T/d + c_2 T/d^{r+1}} B(k, T/d).$$

Therefore, for each $k$ in the above summation, we have that

$$\frac{1}{\sqrt{k(T/d - k)}} \geq \sqrt{(c_1 + c_2 d^{-r})(1 - c_1 - c_2 d^{-r})} d/T.$$

---

[6]A similar technique for $\ell_2$ ball appears in [37] recently, which is also developed independently by [19].

Therefore, the LHS of (35) is lower bounded by

$$C\sqrt{(c_1 + c_2 d^{-r})(1 - c_1 - c_2 d^{-r})}\sqrt{\frac{T}{d}}\frac{2c_2}{d^r} = \Omega(\sqrt{T/d^{2r+1}})$$

for sufficient large $d$. $\qquad\square$

Now we are ready to prove Theorem 5. We choose a particular $\mathbf{x}^T$: We split the $\mathbf{x}^T$ into $d$ blocks each with length of $T/d$. With that, the $i$th part of the inputs and the outputs are denoted by $\mathbf{x}^{(i)} = (\mathbf{x}_{(T/d)*(i-1)+1}, \cdots, \mathbf{x}_{(T/d)*i})$ and $\mathbf{y}^{(i)} = (y_{(T/d)*(i-1)+1}, \cdots, y_{(T/d)*i})$, respectively. We define for any $\mathbf{x}_t$ in the $i$th block $\mathbf{x}^{(i)}$ equals $\mathbf{e}_i$ the standard $d$ base of $\mathbb{R}^d$ with 1 in position $i$ and 0s otherwise. Note that, with these choice of $\mathbf{x}_t$s, we have $\langle \mathbf{w}, \mathbf{x}_t \rangle = w_i$, where $w_i$ is the $i$th coordinate of $\mathbf{w}$ and $\mathbf{x}_t \in \mathbf{x}^{(i)}$.

We will lower bound $r_T^*(\mathcal{H}_f \mid \mathbf{x}^T)$, which will automatically give a lower bound on $r_T^a(\mathcal{H}_f)$. We only need to compute the following Shtarkov sum

$$S_T(\mathcal{H}_f | \mathbf{x}^T) = \sum_{y^T \in \{0,1\}^T} \sup_{\mathbf{w} \in \mathcal{B}_s^d(1)} \prod_{i=1}^{d} P_f(\mathbf{y}^{(i)}|w_i), \tag{36}$$

where $P_f(\mathbf{y}^{(i)}|w_i) = f(w_i)^{k_i}(1 - f(w_i))^{T/d - k_i}$ with $k_i$ being the number of 1s in $\mathbf{y}^{(i)}$. We observe

$$S_T(\mathcal{H}_f | \mathbf{x}^T) \geq \sum_{y^T \in \{0,1\}^T} \prod_{i=1}^{d} \sup_{w_i \in [-d^{-1/s}, d^{-1/s}]} P_f(\mathbf{y}^{(i)}|w_i)$$

$$= \prod_{i=1}^{d} \sum_{\mathbf{y}^{(i)} \in \{0,1\}^{T/d}} \sup_{w_i \in [-d^{-1/s}, d^{-1/s}]} P_f(\mathbf{y}^{(i)}|w_i)$$

$$= \left( \sum_{\mathbf{y} \in \{0,1\}^{T/d}} \sup_{w \in [-d^{-1/s}, d^{-1/s}]} P_f(\mathbf{y}|w) \right)^d$$

$$\geq \left( \sum_{\mathbf{y} \in \{T/d\}} \sup_{w \in [c_1 - c_2 d^{-1/s}, c_1 + c_2 d^{-1/s}]} P(\mathbf{y} \mid w) \right)^d$$

where $P(\mathbf{y} \mid w)$ is as in Lemma 8 and the last inequality holds since $[c_1 - c_2 d^{-1/s}, c_1 + c_2 d^{-1/s}] \subset f([d^{-1/s}, d^{-1/s}])$ by the assumption. Now, Lemma 8 implies that

$$S_T(\mathcal{H}_f \mid \mathbf{x}^T) \geq c^d \left( \frac{T}{d^{(s+2)/s}} \right)^{d/2},$$

where $c$ is some absolute constant that is independent of $d, T$. We conclude

$$r_T^a(\mathcal{H}_f) \geq r_T^*(\mathcal{H}_f) \geq \log S_T(\mathcal{H}_f|\mathbf{x}^T) \geq \frac{d}{2} \log \left( \frac{T}{d^{(s+2)/s}} \right) - O(d)$$

which completes the proof.

# E  Proof of Lemma 5

We first introduce a discretized notion of fat-shattering number, which can be viewed as a misspecified Littlestone dimension [2, 8], see also [25]. For any $\alpha > 0$, we can choose $K \leq \lceil 1/2\alpha \rceil$ points $z_1 < z_2 \cdots < z_K$ in the interval $[0, 1]$ such that any point in $[0, 1]$ is $\alpha$ close to some $z_k$ and $z_{k+1} - z_k = 2\alpha$ for all $k \in [K]$. Now, we define a discretized class $\mathcal{H}'$ for the $[0, 1]$-valued class $\mathcal{H}$ in the following way. For any $h \in \mathcal{H}$, we define function $h' \in \mathcal{H}'$ such that for any $\mathbf{x} \in \mathcal{X}$ we have

$$h'(\mathbf{x}) = \arg\min_{z_k \in \{z_1, \cdots, z_K\}} |z_k - h(\mathbf{x})|,$$

where we break ties arbitrarily.

---

**Algorithm 3** M-SOA algorithm

---

**Input**: Hypothesis class $\mathcal{H}$ with functions map $\mathcal{X} \to [K]$

1: Let $\mathcal{H}^* = \mathcal{H}$
2: **for** $t = 1, \cdots, T$ **do**
3:     Receive feature $\mathbf{x}_t$
4:     For $k \in [K]$, let

$$\mathcal{H}^*_{(\mathbf{x}_t,k)} \overset{\text{def}}{=} \{h \in \mathcal{H}^* \mid h(\mathbf{x}_t) = k\}$$

5:     Make prediction

$$\hat{y}_t = \arg \max_{k \in [K]} \mathbf{FAT}_1(\mathcal{H}^*_{(\mathbf{x}_t,k)})$$

    (where we break ties arbitrarily and deal with empty classes as in Definition 3)
6:     Receive label $y_t$
7:     If $|\hat{y}_t - y_t| \geq 2$, set

$$\mathcal{H}^* = \mathcal{H}^*_{(\mathbf{x}_t,y_t)}$$

8:     If $|\hat{y}_t - y_t| < 2$, set

$$\mathcal{H}^* = \mathcal{H}^*$$

9: **end for**

---

We now view the functions in $\mathcal{H}'$ as functions map $\mathcal{X} \to [K]$ (i.e., we view each $z_k$ as its index $k$). For any discretized class $\mathcal{H}'$, we define the discretized 1-shattering as follows. For any $\mathcal{X}$-valued tree $\tau$ of depth $d$, we say $\mathcal{H}'$ 1-shatters $\tau$, if there exists $[K]$-valued tree s $: \{0,1\}^d_* \to [K]$ such that for any $\epsilon_1^d \in \{0,1\}^d_*$ there exist $h' \in \mathcal{H}'$ such that for all $t \in [d]$:

1. If $\epsilon_t = 0$, then $h'(\tau(\epsilon_1^{t-1})) \leq \mathbf{s}(\epsilon_1^{t-1}) - 1$.
2. if $\epsilon_t = 1$, then $h'(\tau(\epsilon_1^{t-1})) \geq \mathbf{s}(\epsilon_1^{t-1}) + 1$.

**Definition 3.** *The discretized 1-shattering number of a discretized class $\mathcal{H}'$ is defined to be the maximum number $d$ such that $\mathcal{H}'$ 1-shatters some tree $\tau$ of depth $d$. This number is denoted as $\mathbf{FAT}_1(\mathcal{H}')$. If no such tree exists, we define the 1-shattering number to be 0 if $\mathcal{H}'$ is non-empty and $-1$ if $\mathcal{H}'$ is empty.*

The proof of Lemma 5 follows from the following three lemmas.

**Lemma 9.** *The discretized 1-shattering number of $\mathcal{H}'$ is upper bounded by the $\alpha$-fat shattering number of $\mathcal{H}$ where $\mathcal{H}'$ is the discretized class of $\mathcal{H}$ at scale $\alpha$.*

*Proof.* Let $\tau$ be the tree of depth $d$ that is shattered by $\mathcal{H}$ with a $[K]$-valued tee **s**. We define a $[0,1]$-valued tree $\mathbf{s}'$ as follows for any $\epsilon_1^t \in \{0,1\}^d_*$,

$$\mathbf{s}'(\epsilon_1^{t-1}) = z_{\mathbf{s}(\epsilon_1^{t-1})}.$$

We now show that the $\tau$ and $\mathbf{s}'$ are the desired pair that is $\alpha$-shattered by $\mathcal{H}$. This follows from the fact that for any $z_k$ and $z_l$ with $k \neq l$ if some $y \in [0,1]$ is closer to $z_l$, then

$$|y - z_k| \geq \alpha$$

as easy to see. $\qquad \square$

For any discretized class $\mathcal{H}'$, we say a class $\mathcal{G}$ of functions map $\mathcal{X}^* \to [K]$ 1-covers $\mathcal{H}'$ if for any $\mathbf{x}_1, \cdots, \mathbf{x}_T \in \mathcal{X}$ and $h' \in \mathcal{H}'$ there exists $g \in \mathcal{G}$ such that for all $t \in [T]$

$$|h'(\mathbf{x}_t) - g(\mathbf{x}^t)| \leq 1.$$

The following result is crucial for our following analysis, which is an analogy of Lemma 12 of [2] (see also [8, 24]).

**Lemma 10.** *Suppose the discretized 1-shattering number of $\mathcal{H}'$ is upper bounded by $d$, then there exists a 1-covering set $\mathcal{G}$ of $\mathcal{H}'$ such that*

$$|\mathcal{G}| \leq \sum_{t=0}^{d} \binom{T}{t} K^t \leq (TK)^{d+1}.$$

*Proof.* We now describe an algorithm that is similar to the SOA algorithm of [2], which we will call it M-SOA (Algorithm 3)[7]. The algorithm goes as follows: it maintains a running hypothesis class $\mathcal{H}^*$, initially equals $\mathcal{H}'$. Let $(\mathbf{x}_t, y_t)$ be the sample label pair received at round $t$. We will denote by $\mathcal{H}^*_{(\mathbf{x}_t, y_t)}$ the functions in $\mathcal{H}^*$ that is consistent with $(\mathbf{x}_t, y_t)$, i.e., for all $h \in \mathcal{H}^*_{(\mathbf{x}_t, y_t)}$ we have

$$h(\mathbf{x}_t) = y_t.$$

At time step $t$, the algorithm M-SOA will predict $k \in [K]$ such that $\mathbf{FAT}_1(\mathcal{H}^*_{(\mathbf{x}_t, k)})$ is maximum, where we denote by $\mathbf{FAT}_1(\mathcal{H}^*_{(\mathbf{x}_t, k)})$ the discretised 1-shattering number of $\mathcal{H}^*_{(\mathbf{x}_t, k)}$ and break ties arbitrarily. After receiving the true label $y_t$, the M-SOA algorithm will do the following. If $|\hat{y}_t - y_t| \geq 2$, then it sets $\mathcal{H}^* = \mathcal{H}^*_{(\mathbf{x}_t, y_t)}$. Else, it remains on the same $\mathcal{H}^*$. We then continue the prediction procedure for the next time step with the new $\mathcal{H}^*$.

We say the algorithm M-SOA makes an error at time step $t$ if $|\hat{y}_t - y_t| \geq 2$ where $\hat{y}_t$ is the prediction given by M-SOA at time step $t$. We claim that the M-SOA will make at most $d$ errors if the samples $(\mathbf{x}^T, y^T)$ is consistent with some $h \in \mathcal{H}'$.

To see this, we prove by induction on $d$ and $T$ (the base case for $d = 0$ or $T = 0$ is easy to check). Suppose we have observed $\mathbf{x}_1$ at the first step. We show that there can not be two element $k_1, k_2 \in [K]$ such that $|k_1 - k_2| \geq 2$ and both $\mathcal{H}'_{(\mathbf{x}_1, k_1)}$ and $\mathcal{H}'_{(\mathbf{x}_1, k_2)}$ has discretized 1-shattering number $\geq d$. Otherwise, we can concatenate the shattering tree of $\mathcal{H}'_{(\mathbf{x}_1, k_1)}$ and $\mathcal{H}'_{(\mathbf{x}_1, k_2)}$ with the root labeled by $\mathbf{x}_1$ to form a depth $d + 1$ shattering tree of $\mathcal{H}'$ (with $\mathbf{s}(\phi)$ being any number $\in (k_1, k_2)$). This is a contradiction, since the discretized 1-shattering number of $\mathcal{H}'$ is upper bounded by $d$. This shows that either we will make no error at the first step or the discretized 1-shattering number decreased by at least 1 on the remaining consistent class of functions (after $y_1$ has been revealed). For the first case, by induction hypothesis for $T - 1$ we have the number of errors is at most $d$. For the second case, we also have the number of errors upper bounded by $d - 1 + 1 = d$.

We now follow the idea from the proof of Lemma 12 of [2] to construct a covering set $\mathcal{G}$. For any subset $I \subset [T]$ of size $|I| \leq d$ and $\{k_t\}_{t \in I} \in [K]^{|I|}$, we define a function $g$ by running our M-SOA algorithm by changing steps $7 - 8$ as follows. At each time step $t \in [I]$, we update $\mathcal{H}^* = \mathcal{H}^*_{(\mathbf{x}_t, k_t)}$. Otherwise, for any $t \notin I$, we remain on the same $\mathcal{H}^*$. The values of $g$ for each $\mathbf{x}^t$ is given by the output of M-SOA at time step $t$.

Since the M-SOA will make at most $d$ errors if the sample-label pairs $(\mathbf{x}^T, y^T)$ are consistent with some function in $\mathcal{H}'$, we know that any $h \in \mathcal{H}'$ is 1-covered by the function generated by running M-SOA with some $I$ and $\{k_t\}_{t \in I}$ in the above fashion. To complete we observe that by a simple counting argument the number of such pairs $I$ and $\{k_t\}_{t \in I}$ is at most

$$\sum_{t=0}^{d} \binom{T}{t} K^t$$

which completes the proof. □

Finally, we need the following lemma that relates 1-covering of $\mathcal{H}'$ with global sequential $\alpha$-covering of $\mathcal{H}$.

**Lemma 11.** *Suppose there exist a 1-covering set $\mathcal{G}$ of $\mathcal{H}'$, then there exists a global $3\alpha$-covering $\mathcal{G}'$ of $\mathcal{H}$ such that $|\mathcal{G}| = |\mathcal{G}'|$, where $\mathcal{H}'$ is the discretised class of $\mathcal{H}$ at scale $\alpha$.*

*Proof.* For any $g \in \mathcal{G}$, we define a function $g'$ such that for all $\mathbf{x}^t$ we have

$$g'(\mathbf{x}^t) = z_{g(\mathbf{x}^t)}.$$

The claim follows from the fact that any $y$ that is closest to $z_k$ satisfies $|y - z_k| \leq \alpha$ and if some $z$ 1-covers $z_k$ then we have $|z - z_k| \leq 2\alpha$, by triangle inequality

$$|y - z| \leq 3\alpha$$

as needed. □

The proof of Lemma 5 follows from Lemma 9, Lemma 10, and Lemma 11.

---

[7]The major difference with the standard SOA is steps 7-8 and where "M" stands for misspecified.

# F   Proof of Theorem 6

It is sufficient to compute the Shtarkov sum as in (5). For any $y^T \in \{0,1\}^T$ with $k$ 1s, we claim that

$$\sup_{\mathbf{p} \in \mathcal{D}_s} p(y^T) = \frac{1}{k^{k/s}},$$

where

$$p(y^T) = \prod_{t=1}^{T} p_t^{y_t} (1 - p_t)^{1-y_t}.$$

To see this, we use a *perturbation* argument. Denote $I$ be the positions in $y^T$ that takes value 1 such that $|I| = k$. For any $\mathbf{p}$ such that $p(y^T)$ is maximum, we must have $p_j = 0$ for all $j \notin I$. Suppose otherwise, we then can move some probability mass on $p_j$ to some $p_i < 1$ with $i \in I$, which will increase the value of $p(y^T)$, thus a contradiction. Now, we need to show that

$$\prod_{i \in I} p_i \leq \frac{1}{k^{k/s}},$$

this follows easily by AM-GM (i.e., arithmetic mean vs geometric mean) inequality since $\sum_{i \in I} p_i^s \leq 1$ and it takes equality when $p_i = \frac{1}{k^{1/s}}$ for all $i \in I$. Now, the Shtarkov sum can be written as

$$\sum_{k=0}^{T} \binom{T}{k} \frac{1}{k^{k/s}}. \tag{37}$$

To find a lower bound, we only need to estimate the maximum term in the summation. We have

$$\max_k \binom{T}{k} \frac{1}{k^{k/s}} \geq \max_k \frac{T^k}{k^{(1+1/s)k}} \geq e^{\frac{s+1}{s \cdot e} T^{s/s+1}},$$

where the last inequality follows by taking $k = \frac{1}{e} T^{s/s+1}$, and we also use the fact that

$$\binom{T}{k} \geq \frac{T^k}{k^k}.$$

Therefore, we have

$$r_T^*(\mathcal{D}_s) \geq \frac{s+1}{s \cdot e} T^{s/s+1} = \Omega(T^{s/s+1})$$

which completes the proof.