# OpenReview forum: "Precise Regret Bounds for Log-loss via a Truncated Bayesian Algorithm"
_NeurIPS.cc/2022/Conference — NeurIPS 2022 Accept_

### Official Review · Reviewer_75nX · 2022-07-09

**Rating:** 8
**Confidence:** 5
**Soundness:** 4 excellent
**Presentation:** 3 good
**Contribution:** 4 excellent

**Summary:**

The authors consider the problem of sequential prediction of binary data scored with log loss against potentially large classes of reference predictors. Using the method of [RST10] to sequentially construct covers, they algorithmically recover minimax rates that previously were only achieved non-constructively. They also improve constants of (4,4) to (2,1), which they prove are sharp for some classes that have uniform (non-sequential) covers.

**Questions:**

I raised all my questions in the body of the review.

**Limitations:**

I see no negative societal impacts of this theoretical work.

**Strengths And Weaknesses:**

## Review Summary

The main contribution of the work, which is algorithmically achieving the best-known minimax rates for this problem, is a significant contribution to the literature. I have minor comments regarding (a) the authors framing of novelty, as all of the algorithm components exist in the literature already (this is not a bad thing, but warrants stronger citations); and (b) the authors claims of generically tight constants, since it is not clear that the results are even tight in terms of log factors for classes that cannot be uniformly bounded over the covariate space (such as the high-dimensional linear functions example, in contrast with Lipschitz functions on bounded sets). I am confident these concerns can be addressed in the revision period.

## Concerns to be addressed

My first concern is regarding the claims of novelty in this work. The notion of global sequential covering, while concisely defined in Lemma 1, is not novel in my opinion. As the authors cite (unfortunately, only in parentheses and on page 9), this notion was exactly considered by [RST10] in their Section 6.1, who also already proved Lemma 5 (their Lemma 15), albeit with slightly worse constants inside (eventual) log factors. Similarly, the distinction between this notion of sequential covering and the one used for nonconstructive minimax rates (by, e.g., [BFR20]) is only consequential for the algorithmic aspects (which, as noted, is significant and the main contribution of this work is to provide an “implementable” algorithm). This means that Eq. (6) (up to an improvement of constants 4,4 to 2,1) is exactly the same as the main result of [BFR20], since we currently only know how to use this regret bound by further upper bounding it by either the uniform metric entropy (for classes that can be covered without knowledge of covariates) or by the fat-shattering number (for classes that cannot), and both notions of entropy are bounded in the exact same way.

The idea of Bayesian averaging a cover with smoothing has also already been used in the literature to obtain tight rates for log loss regret [see BFR21], and is implicit at least as far back as [YB99] (see the discussion after their Lemma 2); notably, these works are only in the i.i.d. setting, although the smoothing is purely used in the present work to deal with the unboundedness of log loss (the adversarial sequential aspect is handled by the existing cover notion). The truly novel result in the present work is Lemma 4, which allowed the authors to apply these existing notions to the present problem specifically for binary responses. I believe the rhetoric and citations can be updated to reflect this history.

My second concern is regarding the claims of general optimality for the constants. The authors have multiple results of classes for which they prove sharp leading constants. However, for each of these, the class is already uniformly bounded in sup norm over the covariate space, and hence standard notions of covering could be used to control the size, with no need to consider sequential covers (of either kind). For classes that actually require the sequential covering notion, such as the linear classes that require an empirical covering (i.e., it depends on the observed covariates), it is unclear whether even the log factors are tight, let alone the constants. Further, as shown by [BFR20, Corollary 4] (and repeated after Theorem 6 in the present work), sequential fat-shattering (which both [BFR20] and the present work have to use to control their respective covering notions) is insufficient to fully characterize minimax regret. In summary, I think the claim that “our bounds are optimal on the constants” after line 167 is unsubstantiated.

[RST10] - Alexander Rakhlin, Karthik Sridharan, and Ambuj Tewari. Online learning: Random averages, combinatorial parameters, and learnability. Advances in Neural Information Processing Systems, 23, 2010.

[BFR20] - Blair Bilodeau, Dylan Foster, and Daniel Roy. Tight bounds on minimax regret under logarithmic loss via self-concordance. In International Conference on Machine Learning, pages 919–929. PMLR, 2020.

[BFR21] – Bilodeau, Foster, and Roy (2021). Minimax Rates for Conditional Density Estimation via Empirical Entropy, arXiv:2109.10461

[YB99] – Yang and Barron (1999). Information-theoretic determination of minimax rates of convergence, Annals of Statistics.

## Typos

- The definition of regret after line 32 is slightly inconsistent with the usage that follows it, and should be better defined as $R(\phi^T, y^T, \mathcal{H} | x^T)$.
- line 139, “reference class map” is undefined, although reasonably clear; I hope the authors can just be a bit more precise with the wording
- Set notation for sets of functions is somewhat inconsistent, eg write $\mathcal{G} \subseteq [0,1]^{\mathcal{X}^*}$ in Definition 1 to match $\mathcal{H}$

---

> ### Author Response · Authors · 2022-07-28
> **Clarification on statement of novelty**
>
> Thank you very much for the detailed review, helpful comments, and the list of typos.
>
> **Regarding novelty of global sequential covering**: We agree that the same idea was used in [RST10] and also in (Ben-David, Pal, Shalev-Shwartz, 2009). We believe it is worth defining the concept explicitly so that one can study the concept as an independent research object instead of a technical ingredient as implicitly used in the aforementioned papers. We will rephrase our statement of novelty to reflect this.
>
> **Regarding the Bayesian average and smoothing**: Thank you very much for providing the references. We agree that [BFR21] also used the idea of smoothing for controlling the unboundedness of log-loss. We will add the citations and corresponding literature. As noted by the reviewer, these results only consider the case when the covariates are presented i.i.d. More importantly, these results hold only for the average case minimax risk, i.e., the performance of the predictor is compared with the best expert *on average*. While in our paper, we compete with the best experts for any individual $x^T,y^T$.
>
> **Regarding the optimality of constant**: We agree that the tightness of the constants is not substantial for the case where an estimation of the sequential covering number is required since it is not known how to control the log factor. Our main purpose for the statement of optimality on the constant is to emphasize that one should not hope to improve the constants in the form $2 \alpha T+ \log |\mathcal{G}|$ any further. However, it is quite possible that one can obtain other forms of bounds that improve this bound. Moreover, the optimality of the constants allows us to obtain the tightest bound with optimal leading constant for special cases, which may be of independent interest. Indeed, obtaining optimal constants has been long investigated in the information theory and machine learning communities.

---

> > ### Comment · Reviewer_75nX · 2022-08-04
> > **Happy with author responses**
> >
> > Thanks to the authors for their comments. I agree with all of their responses to my review, and have no further questions.

---

### Official Review · Reviewer_Gq3y · 2022-07-11

**Rating:** 7
**Confidence:** 3
**Soundness:** 3 good
**Presentation:** 3 good
**Contribution:** 3 good

**Summary:**

The paper presents new tight lower and upper bounds for online regression with the log loss, with new and simple proofs. For the upper bounds, the authors prove their result using a new and simple truncated Bayesian algorithm. This analysis of this algorithm involves a new complexity measure---the Global sequential covering number---which the author show is upper bounded by the sequential fat-shattering number. For the lower bound, the author use the fact that the sequential minmax regret is lower bounded by the regret in the fixed design setting (known covariates), which they further lower bound using concepts from information theory such as the Shtarkov sum.

**Questions:**

How do your results compare to e.g. "An improper estimator with optimal excess risk in misspecified density estimation and logistic regression"?

**Limitations:**

The presentation can be improved in places. Here I give a couple of suggestions (some minor):
- Line 141: [4, Chapter 3.3] are you referring to Eq 4?
- Lemma 4 should refer to the specific truncation mechanism in Algorithm 2 (not just there exists). This is required in the proof of Theorem 1.
- To be consistent with the notation I would use $(${$0,1$}${}^d)^*$ instead of {$0,1$}${}^d_*$.
- The fact that the global sequential covering number can be bounded by the fat-shattering number should be mentioned earlier in the presentation (perhaps around the definition of the global sequential covering number).

**Strengths And Weaknesses:**

The paper presents improved upper and lower bounds for the problem of online regression with the log-loss. The approaches used to achieve these results are new and simple, which constitute a solid contribution.

---

> ### Author Response · Authors · 2022-07-28
> **Comparison of related papers**
>
> Thank you very much for the detailed review and helpful comments.
>
> **Comparison to "An improper estimator with optimal excess risk in misspecified density estimation and logistic regression"**: The problem considered in that paper is different from the problem we studied in our paper. Specifically, that paper considers the problem of standard supervised regression problem, where the goal is to generate a function $g : X \rightarrow \hat{Y}$ by observing i.i.d. generated samples $X^T, Y^T$ with the goal of minimizing the excessive risk. In our paper, we consider the prediction problem where the samples are generated sequentially, and more importantly, we have no assumption on the statistical mechanism for generating the samples (unlike the i.i.d. assumption in their paper). It is worth noting that a regret bound in our setting automatically implies an excessive risk bound in their setting by using the standard online-to-batch conversion technique. However, as noted in their paper, this conversion may sometimes provide suboptimal bounds in their setting.
>
> **Minor comments**: The citation [4, Chapter 3.3] refers to the ideas used in Theorem 3.2 and Proposition 3.1. However, these results in [4] only study the case for finite experts. Our Lemma 2 is an analog of these results for infinite classes (the proof uses exactly the same idea).

---

### Official Review · Reviewer_K3Wr · 2022-07-12

**Rating:** 7
**Confidence:** 2
**Soundness:** 3 good
**Presentation:** 2 fair
**Contribution:** 3 good

**Summary:**

In this work the authors study bounds on the sequential minimax regret of sequential probability assignment against a set of expert predictors. They provide both lower- and upper-bound on the adversarial regret that either match or improve on previous bounds. For the upper-bounds, their main technique relies on the idea of "sequential $\alpha$-covering" of a experts' class, which builds upon on similar idea from previous work, and using these covers in the classical Bayesian predictor algorithm. They further sharpen such bound when the experts' class is parameterized and Lipschitz over these parameters, and also give sequential covering for experts classes with low fat shattering number. Finally, they also provide interesting lower-bounds on the adversarial regret, which they lower-bound by lower-bounding the maximal minimax regret.

**Questions:**

Unfortunately I do not have interesting technical questions for the authors. However, I'd like to know whether and how the authors plan to improve the presentation of the paper based on what I've discussed. Also, if you could give some intuition on how Shtarkov sums are used to prove lower-bounds, I'd be very interested to hear. However, I know the authors will be time-constrained during the rebuttal and I don't think my review will be a problem for the acceptance of the paper, so feel free to not give a thorough rebuttal to my review.

**Limitations:**

Although the discussion of related work is hard to parse and sometimes limited, the authors do discuss some limitations of their results and future directions of research.

**Strengths And Weaknesses:**

### Strengths
- Although I am not very acquainted with the latest results in regret analysis of sequential probability assignment, this paper seems to have strong theoretical results: better bounds on the minimax sequential regret obtained by interesting ideas combining global sequential coverings (which expands on def of previous work) with Bayesian update rule;
- They also prove lower-bounds that match many of their rates, but I can't say much about the techniques since they do not discuss it at length in the main body of the paper (and I didn't have the time to look into the appendix);

### Weaknesses
- Although I'm making a list, all weaknesses lie under the umbrella of **presentation**. I had a hard time going through this paper due to a collection of reasons (some of which I'll list). In general, the discussion of related work and of the techniques used is sub-optimal and very hard to go through if one doesn't have the other papers fresh in one's mind. A clear example is that only after reading the first couple of pages of [3] (Bilodeau, Foster, Roy 2020) and skimming their results I could digest the paper properly;
- About high-level presentation problems, nowhere in the main body of the paper the authors discuss how they use Shtarkov sums to give lower-bounds. They go through the trouble of even defining it, but only use it on Theorem 1 to prove an upper-bound. Since I do not have time to look through the appendix I'm left wondering why Shtarkov sums are helpful in the l
- For an example of parts where I had trouble reading, I'll walk through my experience in the introduction. The first paragraph of the introduction introduces the problem in a very general (player predicts "labels" instead of probabilities). The second paragraph makes the definition precise and general, and right after already specializes to probability prediction, so it makes one wonder if the generality of labels was even necessary (even more so in the introduction!). Yet, until this point I was still following. When the authors introduce fixed and sequential design, I had to spend quite some time parsing the notation and understanding the differences since the text does not seem to try to guide our understanding. At this point I had to spend more effort than usual just to understand *the introduction* of the paper. But then the "regrets in information theory" section was very dense and even after reading the paper I don't think it helps me place the contribution of the authors in the literature nor does it help me understand the motivation/"hardness" of bounding the regret. In fact, the related work subsection that comes later does a better job (but I'll comment on this later), and I don't know what is the purpose of the section of regret in info theory. Yet, by the end of the introduction I was not sure if I had understood the problem very well, and I definitely could not quite put my finger on what were the bounds that the community had worked on before. That is when I went through [3], whose introduction and defs I could read without much effort, and it helped me understand the motivation and setting much better;
- When comparing your results and techniques with other works, the authors seems to write as if we had these other work fresh in our minds. Even though I do not expect you to explain all the previous work in your paper, more formally stating their bounds or techniques so that I do not need to open their papers to understand what you mean in some passages of your paper would be very helpful. A couple of examples: (1) Before Thm 1 you mention something about trees in [23], but without opening the paper a reader cannot even understand why it is a problem or what it means. After Thm 1 you also say that you obtain better constants if compared to [3], but without opening their paper I do not know how much better theses constants are and whether they analyze a more general case. At least stating the constants [3] gets should not take much space; (2) In example 1 it is not immediately clear what constants are being discussed (although one can infer which constants after reading the rest of the paper) and the authors also mention  logarithmic dependency on $L$, but they never mention what were the dependency on $L$ on previous work (was it better? worse?)


Some briefer comments on presentation:
- The discussion between lines 167 and 170 should be a proposition, and the authors could be more explicit on what "universally" means;
- The definition of Big-Oh notation in lines 120 and 121 is a bit wrong (it is not for all $t \geq 0$, only for sufficiently big $t$);
- "function is self-concordance" -> "function is self-concordant"
- Lines 205-210 are extremely hard to parse (and formatted weirdly);
- Line 213: cardinally -> cardinality
- In line 214 you cite [23] but it only made me confused: is it that the next lemma is due to them, or they do something related?

### Summary
I think this is a technically strong paper which is very hard to parse for readers not well-acquainted with the latest work on this topic. I'm favorable towards acceptance. For now I only have my score in "weak accept" because I have a hard time assessing impact (even after skimming a bit of related work), but I might increase my assessment during the rebuttal phase.


### Post-rebuttal
Increase my score from 6 to 7 based on other reviews and authors responses. This is definitely a strong paper and my initial problems with presentation will be addressed as much as possible.

---

> ### Author Response · Authors · 2022-07-28
> **Clarification on the presentation**
>
> Thank you very much for the detailed review. We apologize for the lack of clarity in our presentation. We note that a nine-page limit significantly inhibits our ability to present background and related results in a comprehensive and self-contained manner.  We provide a brief overview of the literature below that may be more accessible for readers who are not intimately familiar with related work. We are happy to include it in the final manuscript.
>
> **Overview:** Sequential probability assignment under logarithmic loss is a fundamental problem that has been extensively studied in the information theory community, due to its close connection to universal compression. However, information theory only considers *simulatable* experts, i.e., experts that make predictions based only on previously observed labels (there are no features, $\textbf{x}^T$). It can be shown that the regrets for simulatable experts are completely characterized by the Shtarkov sum. In (Rakhlin and Sridharan, 2015), the authors made an important extension of the framework to allow experts to make predictions that may depend on some side information (i.e., the features, $\textbf{x}^T$). Their main techniques are the concept of (local) sequential covering and chaining. However, their approach only works when the losses are Lipschitz and bounded, which does not apply to log-loss. To deal with this issue, they used a hard truncation approach that truncates any function $h$ to $h'$ such that $h'(x) = \delta*1(h(x) < \delta) + h(x)*1(\delta \le h(x) \le 1-\delta) + (1-\delta)*1(h(x)> 1-\delta)$. This approach could reduce the gradient of log-loss from unbounded to bounded. However, it still scales as $1/\delta$, where $\delta$ is an additional parameter and is the source of suboptimality of their approach. (Bilodeau, Foster, Roy 2020) observed that by exploiting the self-concordancy of log-loss, one can bypass the truncation approach and obtain tighter bounds that are tight for specific classes (i.e., they can not be improved universally). However, their approach is non-constructive. Despite the non-triviality of the proof of (Bilodeau, Foster, Roy 2020), we showed in our paper, perhaps surprisingly, that their bounds can be achieved algorithmically (i.e., with an implementable algorithm) by using a simple smooth truncation approach (cf. Lemma 4), which is different from the truncation of (Rakhlin and Sridharan, 2015). Moreover, our bounds improve the constants of (Bilodeau, Foster, Roy 2020) from (4,4) to (2,1), which are tight for specific classes (i.e., the constants are not universally improvable). Our approach also uses the concept of global sequential covering, which was implicitly used in (Rakhlin, Sridharan and Tewari, 2010) and dates back to the ideas in (Ben-David, Pal, Shalev-Shwartz, 2009).
>
> **For the  lower bounds based on Shtarkov sum**: The main observation is that when we have features $\textbf{x}^T$ known in advance, the fixed design regret introduced in our paper is equivalent to the classical *simulatable* case, and more importantly, fixed design regrets are always lower bounds for the sequential regret. One can therefore analyze fixed design regret using Shtarkov sum by choosing some hard $\textbf{x}^T$ (the selection of $\textbf{x}^T$ is the main technical part). See, Theorem 5 and 6 for examples of how appropriately selected $\textbf{x}^T$ leads to tight lower bounds.
>
> **Other comments**:
> * The dependency on the Lipschitz constant $L$ was considered by [11] where they have linear dependency on $L$, though their Lipschitz condition is on $\log f$ instead of $f$. [3] did not explicitly mention the dependency on $L$, however, their result can also provide a logarithmic dependency but with a worse leading constant.
> * In line 214, the Lemma 5 is ensentially the results esteblished in [23, Section 6.1], by combining their Lemma 14 and 15.

---

> > ### Comment · Reviewer_K3Wr · 2022-08-05
> > **No further questions and increase in score**
> >
> > I'd like to thank the authors to carefully address my concerns. I was the only reviewer that had a hard time with the presentation, so do not feel obligated to drastically change the paper because of it. My guess is that I am more acquainted with the study of (simulatable) experts in the online learning literature, and the language used made it hard for me to find my footing. Yet, since NeurIPS attendees have wide-ranging interests, having a more accessible introduction and presentation is always welcome. The overview you gave is quit helpful and I enjoyed the summary on the techniques used in the lower-bound.
> >
> > Finally, the other reviews, mainly from 75nX, helped me better understand the context of the contributions. I will increase my score to reflect my confidence that the authors will improve the presentation as much as space allows and to reflect my improved understanding of their contributions.

---

### Meta-Review · Area_Chair_Fj9b · 2022-08-22

**Recommendation:** Accept
**Confidence:** Certain

**Metareview:**

This paper considers the problem of online learning with the logarithmic loss, and provides a new algorithm based on smoothing of the log loss which matches certain rates that were previously only achieved through non-constructive methods.

Reviewers agreed that the algorithm and proof technique are novel, and that the resulting regret bounds improve over the state of the art. For the final version of the paper, the authors are encouraged to incorporate the reviewers' comments regarding presentation.

**Award:**

No

---

### Decision · Program_Chairs · 2022-09-14

Accept